# Effect of Thermal Pretreatments on Phosphorylation of *Corypha umbraculifera* L. Stem Pith Starch: A Comparative Study Using Dry-Heat, Heat-Moisture and Autoclave Treatments

**DOI:** 10.3390/polym13213855

**Published:** 2021-11-08

**Authors:** Basheer Aaliya, Kappat Valiyapeediyekkal Sunooj, Chillapalli Babu Sri Rajkumar, Muhammed Navaf, Plachikkattu Parambil Akhila, Cherakkathodi Sudheesh, Johnsy George, Maximilian Lackner

**Affiliations:** 1Department of Food Science and Technology, Pondicherry University, Puducherry 605014, India; aaliyab2647@gmail.com (B.A.); china.rajkumar7@gmail.com (C.B.S.R.); navafmnk@gmail.com (M.N.); akhilapp2018@gmail.com (P.P.A.); sudheeshsuthi8@gmail.com (C.S.); 2Food Engineering and Packaging Division, Defence Food Research Laboratory, Mysore 570011, India; g.johnsy@gmail.com; 3Department Industrial Engineering, University of Applied Sciences Technikum Wien, Höchstädtplatz 6, 1200 Vienna, Austria

**Keywords:** talipot starch, non-conventional starch, dry-heat treatment, heat-moisture treatment, autoclave treatment, chemical modification, crosslinking, phosphodiester bond, resistant starch

## Abstract

Talipot starch, a non-conventional starch source with a high yield (76%) from the stem pith of talipot palm (*Corypha umbraculifera* L.) was subjected to three different thermal treatments (dry-heat, heat-moisture and autoclave treatments) prior to phosphorylation. Upon dual modification of starch with thermal treatments and phosphorylation, the phosphorous content and degree of crosslinking significantly increased (*p* ≤ 0.05) and was confirmed by the increased peak intensity of P=O and P–O–C stretching vibrations compared to phosphorylated talipot starch in the FT-IR spectrum. The highest degree of crosslinking (0.00418) was observed in the autoclave pretreated phosphorylated talipot starch sample. Thermal pretreatment remarkably changed the granule morphology by creating fissures and grooves. The amylose content and relative crystallinity of all phosphorylated talipot starches significantly decreased (*p* ≤ 0.05) due to crosslinking by the formation of phosphodiester bonds, reducing the swelling power of dual-modified starches. Among all modified starches, dry-heat pretreated phosphorylated starch gel showed an improved light transmittance value of 28.4%, indicating reduced retrogradation tendency. Pasting and rheological properties represented that the thermal pretreated phosphorylated starch formed stronger gels that improved thermal and shear resistance. Autoclave treatment before phosphorylation of talipot starch showed the highest resistant starch content of 48.08%.

## 1. Introduction

Starch is a naturally occurring biopolymer with an extended application in food and non-food industries, owing to its technological properties, bioavailability, biodegradability, and safety. It is a well-favored, multifunctional and renewable material majorly sourced from cereals, seeds, roots, and tubers [1]. Starch is a storage polysaccharide made up of chains of linear amylose and branched amylopectin. It is considered to be a versatile food ingredient used as a thickener, binder, emulsifier, texturizer, and gelling agent [2]. However, native starch possesses certain undesirable characteristics such as low shear stress resistance, high retrogradation and syneresis tendency [3,4]. Hence, native starch is subjected to physical, chemical, enzymatic, and genetic modifications to improve functionality and nutritional benefits, such as increasing resistant starch (RS) content [5].

Physical modification of starch is further classified into thermal and non-thermal modifications. Among thermal modifications, dry-heat treatment (DHT), heat-moisture treatment (HMT), annealing, and autoclave treatments are most commonly used for starch modification. DHT is a thermal treatment of starch at high temperatures (130–140 °C) for a duration of 1–20 h [6]. DHT changes the morphological and physicochemical attributes and alters the thermal, rheological and pasting properties of starch. DHT modifies the protein functionality and Maillard reaction and enhances the solubility, oil-binding capacity, and gel firmness of modified starches. Dry-heat-treated flour can be employed as a baking improver and to enhance the Maillard reaction by partial substitution with wheat flour [7]. In addition, DHT starches make stronger gels with reduced syneresis that can be used for 3D food printing applications [8]. HMT is a hydrothermal modification where the starch is treated in a moisture content of less than 35% *w*/*w* at a temperature ranging from 80–140 °C for a duration of 1–24 h [9]. The significant changes in the functionality of heat-moisture treated starch are due to the increased starch chain interactions that lead to a rupture of crystalline structure and a disruption of the double-helix structure, which subsequently rearrange themselves [10]. HMT majorly affects granule morphology and crystallinity, double helix content, amylose-lipid complexes, in vitro digestibility, and pasting and gel properties of starches [9]. Heat-moisture treated starches with enhanced thermal and shear stability are used in noodle and canned food processing industries [11]. Autoclaving is widely performed for assuring food safety in industries. Autoclave treatment is applied for modifying starch at high temperatures with a pressure range of 3–3.5 bar in the presence of moisture for a specific period [12]. The process of autoclaving executes compression and decompression cycles for facilitating the interaction of water vapor with starch [6] and enhances the hydration of amorphous regions of starch granules under a pressure field, leading to the rearrangement of double helical structure of amylopectin [12]. Autoclave treatment disorganizes starch granules that lead to increased availability of the amylose and amylopectin chains to develop more inter- or intra-associations based on starch type, autoclave pressure and temperature. Autoclave treatment remarkably influences the digestibility of starch by RS formation by the newly formed starch chain associations [13].

Phosphorylation is an established food-grade chemical treatment employed for modifying starch of varied sources [14]. Phosphorylation by crosslinking forms intra- and inter-molecular bonds between hydroxyl groups of starch molecules in the amorphous region and improves starch paste stability [15]. Upon crosslinking, the new chemical bond (phosphodiester bond) developed enhances the shear, heat and acid stability of starches [14]. Phosphorylation alters the properties of modified starches by lowering solubility and swelling power and enhancing gelatinization temperature and RS content of starch granules [16]. However, the extent of phosphorylation is dependent on starch source, crosslinking agent, and degree of crosslinking (DC). Phosphoryl chloride, epichlorohydrin, monosodium orthophosphate, sodium tripolyphosphate, and sodium trimetaphosphate are the chemical reagents used for phosphorylation [16]. Upon phosphorylation, the establishment of orthophosphate groups into a starch structure generates mono- or di-starch phosphates, which are potentially approved as additives (E1410, E1412, E1413, E1414, and E1442) in food industries by the European Union Commission Regulation 1129/2011 at quantum satis level [5].

From the literature thus far, it is evident that dual modification is observed to make remarkable alterations in starch characteristics compared to single modifications. For instance, dual modification of waxy maize starch with crosslinking and HMT is reported to change the physicochemical properties, in vitro digestibility and increase the RS content of modified starches [17]. Besides, starch crosslinking has also been dual modified with annealing, acetylation, and hydroxylpropylation. However, limited works detail the combined effect of phosphorylation and thermal treatments in starches. This is the first report to explore and compare three thermal modification (DHT, HMT, and autoclave treatment) effects on phosphorylation of a non-conventional starch from talipot palm. Talipot starch of A-type crystallinity is an underexploited starch from the stem pith of talipot palm (*Corypha umbraculifera* L.). Talipot palm is a large hapaxanthic palm from the family of *Arecaceae* that has the largest terminal paniculate inflorescence in the entire flora [18]. Talipot palm is native to Sri Lanka, Myanmar, Malaysia, and India. Approximately 76% of starch yield is obtained from talipot stem pith flour, and the talipot starch possesses about 28% amylose [2]. The objective of the present study is to examine the impact of thermal pretreatments on the phosphorylation of talipot starch for determining its potential application in food industries.

## 2. Materials and Methods

### 2.1. Materials

The talipot flour was acquired from the stem pith of a matured talipot palm from Malappuram, South India, by following the method of Navaf et al. [2] and stored in a refrigerator (4 ± 2 °C). Phosphoryl chloride (POCl_3_, >99% purity), potassium iodide, α-amylase from *A. oryzae* (∼30 U/mg), amyloglucosidase from *A. niger* (≥300 U/mL) were procured from Sigma Aldrich (St. Louis, MO, USA). Sodium hydroxide, hydrochloric acid, acetic acid, and iodine were purchased from Merck Ltd. (Mumbai, India). The chemicals availed for the study were of analytical grade, otherwise mentioned. 

### 2.2. Isolation of Talipot Starch

Talipot starch was isolated by adhering to the procedure of Sudheesh et al. [19] with necessary modifications. The flour was taken in a 1:9 ratio with distilled water (DW), and the pH of the solution was changed to 10–12 with 0.05 M NaOH. The suspension was stirred at 1000 rpm for 3 h and filtered in a 100-mesh test sieve (Jayant Scientific Industries, Mumbai, India). The pH of the suspension was brought to neutral with 0.1 M HCl, followed by centrifugation at 1500 rpm for 15 min to obtain the light brownish color sediment of starch. The centrifugation was repeated until complete removal of dark-brown colored residues formed as a top layer along with supernatant. The obtained starch sediment was dried overnight in a hot air oven at 40 °C. The dried starch was finely ground, sieved, and stored in an air-tight container under refrigeration conditions (4 ± 2 °C) for modifications. The native talipot starch was considered to be the control and was referred to as NTS.

### 2.3. Phosphorylation of NTS

Talipot starch was phosphorylated by following the method of Hazarika and Sit [20] with necessary modification. Starch (30 g) was dispersed in 48 mL DW and stirred at 25 °C along with 0.6 g Na_2_SO_4_. The pH of the suspension was adjusted to 11.0 using 1 M NaOH and POCl_3_ (0.10%, *v*/*w*) was incorporated dropwise into the suspension. The starch was subjected to phosphorylation reaction for 1 h with continuous stirring at 200 rpm by maintaining the pH at 11.0. Later, the reaction was ceased by adjusting the pH of the slurry to 6.5 using 0.1 N HCl. The resultant mixture was centrifuged (3000 rpm, 10 min) thrice with DW and once with absolute ethanol. The obtained starch sediment was then dried overnight in a hot air oven at 40 °C and finely ground, sieved, and stored for analysis. The phosphorylated talipot starch was referred to as PTS.

### 2.4. Thermal Modifications of NTS

DHT of native talipot starch was performed by following the method of Maniglia et al. [8], where the evenly spread starch (approx. 1 mm thickness) was covered with an aluminum foil and secured with high-temperature tape to prevent the loss of moisture. The packed starch sample was then treated in a hot air oven at 130 °C for 4 h.

HMT was performed by following the method of Marboh and Mahanta [21] with slight modification. The moisture content of talipot starch was made to 30% with DW, equilibrated for 24 h at 4 ± 2 °C and heat-moisture treated for 12 h at 110 °C.

Autoclave treatment was performed according to the method described by Deka and Sit [22] with slight modification, where the moisture content of talipot starch was made to 30% and autoclaved at 121 °C for 45 min. The starch sediments after thermal modifications were dried overnight in a hot air oven at 40 °C, finely ground and sieved. The thermally modified talipot starch samples by DHT, HMT and autoclave treatment was referred as DHTS, HMTS and ACTS.

### 2.5. Pretreatment of PTS with Thermal Treatments

Talipot starch samples thermally treated by DHT, HMT, and autoclave treatment were then subjected to phosphorylation (as mentioned in 2.3). The resultant mixtures were centrifuged (3000 rpm, 10 min) thrice with DW and once with absolute ethanol. The obtained starch sediments were then dried overnight in a hot air oven at 40 °C and finely ground, sieved, and stored for further analysis. The thermal pretreated phosphorylated talipot starch by DHT, HMT, and autoclave treatment was referred to as DH-PTS, HM-PTS, and AC-PTS, respectively.

### 2.6. Phosphorous Content and Degree of Crosslinking

The phosphorus content (P) of the native and modified starches were quantified according to the spectrophotometric method of Smith and Caruso [23], and the degree of crosslinking (DC) was determined with the formula [15]:(1)$$\mathrm{DC}=\frac{162\;P}{\left(3100-102\;P\right)}$$

### 2.7. Amylose Content

The rapid colorimetric method of Williams et al. [24] was adopted to estimate the amylose content of native and modified talipot starches. The absorbance was determined with the help of a UV/visible scanning spectrophotometer (Shimadzu, UV-1800, Kyoto, Japan) at 620 nm.

### 2.8. SEM

The morphological attributes of native and modified talipot starch samples were determined using scanning electron microscopy (SEM) (HITACHI, S-3400N, Tokyo, Japan). Each sample was mounted over a stub using double-sided carbon conductivity tape and carbon-coated in an automated sputter coater (HITACHI, E-1010, Tokyo, Japan). The coated talipot starch samples were viewed at 1500× magnification.

### 2.9. XRD and RC

The crystalline characteristics of native and modified talipot starch samples were analyzed under an X-ray diffractometer (XRD) (BRUKER, D2 PHASER, Karlsruhe, Germany). The samples were molded flat onto the cavity of a sample holder, and the data on scanning was obtained at an angle of diffraction (2θ) = 4° to 60°. The relative crystallinity (RC) of the starch granules was estimated with the formula:(2)$$\mathrm{RC}\;\left(\%\right)=\frac{\mathrm{Crystalline}\;\mathrm{Area}}{\mathrm{Total}\;\mathrm{Area}}\times 100$$

### 2.10. FT-IR

FT-IR spectra of native and modified talipot starch samples were studied using an FT-IR spectrophotometer (Thermo Fisher Scientific, Nicolet 6700, Waltham, MA, USA). Each starch sample was blended with optical grade KBr in a ratio of 1:100 and made into pellets using a pelletizer (Shimadzu, MHP-1, Kyoto, Japan). The FT-IR analysis was performed in the scanning range, 400–4000 cm^−1^.

### 2.11. Swelling Power and Solubility

The swelling power and solubility of starch samples were analyzed by following the procedure detailed elsewhere [25] and calculated with the formulae:(3)$$\mathrm{Swelling}\;\mathrm{power}\;\left(\frac{g}{g}\right)=\frac{\mathrm{Wt}.\;\mathrm{of}\;\mathrm{swollen}\;\mathrm{granules}}{\mathrm{Wt}.\;\mathrm{of}\;\mathrm{sample}\;}$$
(4)$$\mathrm{Solubility}\;\left(\%\right)=\frac{\mathrm{Wt}.\;\mathrm{of}\;\mathrm{dried}\;\mathrm{supernatant}}{\mathrm{Wt}.\;\mathrm{of}\;\mathrm{sample}}\times 100$$

### 2.12. Light Transmittance

Light transmittance was found by employing the method described in Navaf et al. [26] at 640 nm.

### 2.13. Pasting Properties

The pasting properties were found with the help of Rapid Visco Analyzer (Newport Scientific, RVA StarchMaster 2, Warriewood, Australia). A thoroughly mixed 12% starch slurry of each sample was taken in a canister. The slurry was equilibrated for 1 min at 50 °C, heated for 3.5 min at 95 °C, and held for 2.5 min at 95 °C. The temperature was cooled down to 50 °C for 4 min and held at 1 min. The parameters of pasting temperature (PT), peak viscosity (PV), breakdown viscosity (BDV), final viscosity (FV), and setback viscosity (SBV) were analyzed.

### 2.14. Textural Properties

Textural properties of native and modified talipot starch gels were obtained using texture analyzer (TA.HD*plusC*, Stable Micro Systems, Surrey, UK). The gelatinized starch samples in canisters after RVA analysis was stored at 20 ± 2 °C for 24 h by covering with parafilm. The starch gels were punctured at 1.0 mm/s to 30 mm distance using a stainless-steel punch probe (p/5 probe of 5 mm diameter). Textural properties such as hardness, adhesiveness, springiness, cohesiveness, and gumminess were determined from the curve between force and time. 

### 2.15. Dynamic Rheology

The storage (G′) and loss (G″) moduli and loss factor (tan δ) of native and modified talipot starch gels were analyzed with the method described in Sudheesh et al. [27] at 25 °C with a rotational rheometer (Anton Paar GmbH, RheolabQC, Graz, Austria) having a titanium cone CP75-1 with an angle of 1° plate geometry possessing 0.07 mm slit and 40 mm measurement. The starch slurry of 6% was gelatinized and cooled to room temperature to prepare starch gel.

### 2.16. In Vitro Digestibility

Rapidly digestible starch (RDS), slowly digestible starch (SDS), and resistant starch (RS) were determined by following the method of Englyst et al. [28].

### 2.17. Statistical Analysis

The data reported were the average of triplicate observations and conveyed as mean ± standard deviation (SD), at a significant difference of *p* ≤ 0.05 of the starch samples, NTS, PTS, DHTS, HMTS, ACTS, DH-PTS, HM-PTS and AC-PTS. The data from each experiment were analyzed with one-way analysis of variance (ANOVA) and Duncan’s multiple range test (DMRT) employing the software, IBM SPSS Statistics 23 (IBM Corporation, Chicago, IL, USA).

## 3. Result and Discussions

### 3.1. Degree of Crosslinking and Amylose Content

Phosphorus content (P) is determined for interpreting the degree of crosslinking (DC) upon phosphorylation with any crosslinking agent [14]. The phosphorus increased upon crosslinking is determined by the subtraction of endogenous phosphorus present in native starches. The P of NTS was found to be 0.018%, and it is comparable to the P of native corn starch (0.02%) [15]. Thermal treatment did not significantly (*p* > 0.05) change the P of talipot starches. Upon phosphorylation with POCl_3_, the P content of talipot starch increased from 0.018% to 0.058%. The introduction of orthophosphate groups into starch structure upon crosslinking increased the P content. Besides, when talipot starches were subjected to thermal pretreatment prior to phosphorylation, the P content and DC significantly increased (*p* ≤ 0.05) compared to P content and DC of PTS sample, where the highest P content (0.080%) and DC (0.00418) was shown by AC-PTS sample (Table 1). The realignment of starch granules after hydrothermal treatment under pressure paved the way for increased bond formation and DC upon phosphorylation.

Amylose renders a prime role in governing starch gelling and pasting properties. NTS possessed a higher amylose content of 28.81%, and it significantly decreased (*p* ≤ 0.05) upon modifications, except for ACTS (Table 1). The decrease in amylose content upon phosphorylation can be ascribed to the phosphodiester bond formation between the starch chains that leads to the reduced rate of the iodine-amylose complex formation [20]. Besides, higher alkaline pH established during phosphorylation might have leached out amylose chains from starch granules, reducing the amylose content. Crosslinked taro starch with 0.05% and 0.10% POCl_3_ exhibited a similar decrease in amylose content [20]. The decrease in amylose content of HMTS might be due to some extent of partial hydrolytic effect and amylose leaching after HMT [10]. The breakdown of amylopectin chains by the effects of high pressure and temperature by autoclave treatment led to the formation of linear amylose chains, which subsequently increased the amylose content in ACTS sample. A consistent result was observed in a previous study of Deka and Sit [22]. Thermally pretreated phosphorylated starches showed significantly reduced (*p* ≤ 0.05) amylose content than PTS, where DH-PTS showed the lowest amylose content (19.26%). High-temperature treatment followed by phosphorylation introduced phosphodiester bonds and enhanced interaction between amylose-amylopectin, making amylose more insoluble and inaccessible for binding with iodine during quantification [10].

### 3.2. Morphology Analysis

The morphological attributes of native and modified talipot starch samples were analyzed by SEM and are provided in Figure 1. The SEM monograph of NTS showed that irregularly shaped starch granules possessed a smooth surface. The NTS granules were either round, elongated, or elliptical shapes of varying sizes. The length and breadth of talipot starch granules were found to be in the range of 13.75–62.21 and 12.50–30.62 µm, respectively. However, a remarkable change in surface characteristics was noticed after modifications. PTS granules showed roughness and surface erosion, which might be contributed by the exposure of starch granules to a prolonged alkaline medium rendered by sodium hydroxide during phosphorylation treatment [20]. Similar morphological changes were exhibited by crosslinked wheat and taro starches [20,29]. Thermal treatments with DHT, HMT and autoclave treatment created fissures on the surfaces of the DHTS, HMTS and ACTS granules. Conversely, DHT, HMT and autoclave treatments followed by phosphorylation imparted more pronounced morphological changes in talipot starch granules. Fissures, grooves, and surface erosion was noticed in dual-modified talipot starches. The high temperature in DHT accelerated the movement of the molecules and created cracks and grooves in the DH-PTS sample [30]. HMT and autoclave treatment in the presence of increased moisture content and thermal force made the starch granules actively absorb more moisture, remarkably changing the granule morphology [31].

### 3.3. XRD and RC Analysis

On the basis of amylopectin branched-chain packaging in the double helix, the crystallinity pattern of starch is determined by XRD analysis. NTS exhibited a minor peak at 2θ, 11.2° and major peaks at 15.1°, 17.2°, 18.1° and 23.2° (Figure 2a). The crystalline pattern of NTS exhibited an A-type polymorphism, and it did not change with single and dual modifications, suggesting that no effect on the crystalline pattern was observed upon phosphorylation and thermal treatments.

The RC of NTS was observed to be 16.97%, and PTS showed an insignificant decrease (*p* ≥ 0.05) in RC upon phosphorylation treatment (Table 1). This unremarkable change in crystallinity indicates that phosphorylation mainly occurs in amorphous regions [17]. However, crosslinked faba bean, yellow field pea, and corn starch treated with POCl_3_ (1 and 2%, *v*/*w*, sb) exhibited a significant decrease (*p* ≤ 0.05) in RC. This might be ascribed by the substitution of some hydroxyl groups with phosphate group during phosphorylation [15]. However, thermal treatments significantly decreased (*p* ≥ 0.05) the RC of DHTS, HMTS and ACTS. The decrease in the RC of DHTS may be attributed to the degradation of crystalline regions of starch, partial gelatinization of starch granules, and double helical movement during DHT [32]. HMT and autoclave treatment disrupted the organized the crystalline structure and also resulted in the imbalance of the lamellar matrix, decreasing RC [10].When thermal pretreatment was performed prior to phosphorylation, the RC significantly decreased (*p* ≤ 0.05) compared to PTS, and the lowest RC (10.60%) was observed in the DH-PTS sample. This decrease is due to enhanced degradation of starch crystallites or alteration in the orientation of crystallites [32], despite the recovery effect imparted by crosslinking via phosphorylation for crystalline disruption. Consistent reduction in RC of proso millet starch and red adzuki bean starch upon DHT were reported elsewhere [32,33]. The decrease in RC of AC-PTS was attributed to the disruption of organized crystalline structures and reduction in crystalline perfection of talipot starch granules by the action of steam under pressure during autoclave treatment [22].

### 3.4. FT-IR Spectral Analysis

FT-IR spectra represent the stretching, bending, and deformation bands corresponding to the functional groups present in starch granules. The FT-IR spectra of native and modified talipot starches are shown in Figure 2b. A broad peak at 3438 cm^−1^ illustrates the O–H group stretching vibrations, and the peak noticed at 2927 cm^−1^ by a stretching vibration shows the presence of the CH_2_ group [2]. The peak observed at 1640 cm^−1^ indicated the H–O–H bending vibration of water molecules found in the amorphous regions. The peaks noticed at 1444 and 989 cm^−1^ shows CH_3_ and C–O–H bending vibrations, respectively [26]. The increased peak intensity of all phosphorylated talipot starches at 1160 and 1010 cm^−1^ depicts P = O and P–O–C stretching vibrations, respectively. Compared to NTS, the increase in the intensity of these two peaks confirms the phosphorylation reaction with the formation of phosphate crosslinks in the modified starches [16]. Thermal modification on DHTS, HMTS and ACTS samples did not add a new characteristic peak to the FT-IR spectra; however, noticeable change in peak intensity indicating O–H groups was observed in the thermally modified samples. With thermal pretreatment, the peak intensity depicting P = O and P–O–C stretching vibrations of DH-PTS, HM-PTS, and AC-PTS samples increased compared to PTS, which can be correlated with the increased P content and DC of dual-modified talipot starches.

### 3.5. Swelling Power and Solubility

The interaction of the amorphous and crystalline region of the starch chain with water molecules is indicated by swelling power and solubility. The swelling power of native and modified talipot starches increased with the increase in temperature from 60 to 90 °C (Figure 3a). The augmentation in swelling power of starch was caused by the breaking of amylose and amylopectin present in starch granules, which destructs the crystalline regions and causes swelling of starch granules [2]. The swelling index of single and dual-modified samples decreased significantly (*p* ≤ 0.05), where the least swelling power was observed in the AC-PTS sample. In the PTS sample, the reduction in swelling power might be attributed to the strengthened intermolecular interaction by the introduction of phosphodiester bonds upon crosslinking. The addition of phosphate groups thus tightly held the starch granules and formed a gelatinized mass that inhibited swelling even at higher temperatures [16]. Phosphorylated starch from sago and taro also indicated reduced swelling power in previous studies [16,20].

The swelling power of thermally modified samples was significantly lower (*p* ≤ 0.05) than NTS. The reduced swelling power of DHTS was attributed to the extent association between starch chains, which caused further reduction in the diffusion of amylopectin molecules [30]. The decreased granular stability that leads to unravelling of double helices in starch granules upon hydrothermal treatment reduced the swelling power of the HMTS sample [11]. The development of strong bonding forces between granules by thermal treatments and phosphorylation played a major role in limiting the swelling power of DH-PTS, HM-PTS, and AC-PTS. The AC-PTS sample exhibited the lowest swelling power, and it ranged from 1.34–4.24%. The enhanced intramolecular bonding thereby decreased the absorption and retention ability of amylopectin and reduced the hydration capacity of the starch granules [34]. Hence, thermal treatments and phosphorylation synergistically reduced the swelling power of dual-modified talipot starch samples. Dual-modified waxy maize starch by HMT and crosslinking (STMP/STPP, 99:1 (5, 10, and 12%, *w*/*w*)) represented a similar decrease in swelling power [17].

Starch solubility is associated with the amylose leaching from the granules, followed by maximum swelling. The solubility of native and modified talipot starch samples showed a significant increase (*p* ≤ 0.05), with an increase in temperature from 60 to 90 °C (Figure 3b). DHTS showed the highest solubility value (7.50%) at 90 °C. High temperature may cause the complete migration of amylose from the surface of amylopectin crystals and increase the leaching of amylose, resulting in higher solubility [32]. However, the decrease in solubility after hydrothermal is due to the internal rearrangement of starch granules, providing enhanced interactions between starch functional groups and development of amylose-lipid complexes within starch granule [35]. The solubility of talipot starch decreased significantly (*p* ≤ 0.05) upon phosphorylation owing to the restricted leach out of starch chains due to the formation of phosphodiester bonds. Such formed crosslinks protected the structure of starch granules and hindered the disintegration and leaching out of starch molecules [29]. The solubility of HM-PTS and AC-PTS samples was further reduced and were significantly lower (*p* ≤ 0.05) than PTS. The decrease was due to the formation of crosslinks between starch chains that helps in the stabilization of readily soluble amylopectin presented in the amorphous regions [17]. Consistent results were observed when wheat starch was modified by crosslinking with POCl_3_ and dual modified by crosslinking and organic acids [29]. However, the solubility of DH-PTS was significantly higher (*p* ≤ 0.05) than PTS. The increase in DH-PTS solubility might be attributed to the damage in the amorphous region as well as in the granule surface that led to amylose leaching and increased starch solubility during DHT [30], which was not able to overcome by further treatment with phosphorylation. This demonstrated that the disrupted DH-PTS starch granules were prone to swell and to absorb water. Dry-heat-treated red adzuki bean starch also indicated decreased solubility in the previous study [32].

### 3.6. Light Transmittance

Higher starch paste clarity is an important characteristic in the preparation of certain products such as fruit pastes, jellies, and pie fillings that require transparency, and it is indicated by a higher light transmittance value [20]. The light transmittance value of native and modified talipot starch samples decreased as the storage period increased due to the turbidity formation in starch gel, as shown in Figure 4. This is because of the reassociation of leached amylose–amylose and amylose–amylopectin chains by retrogradation [2]. Thermal modification with DHT significantly increased (*p* ≤ 0.05) and HMT and autoclave treatment significantly decreased (*p* ≤ 0.05) the light transmittance of talipot starches. The lower light transmittance value of hydrothermally modified starches was attributed to the increase in flexibility of starch chains within the amorphous region of granules [35]. The process of phosphorylation remarkably decreased the paste clarity of modified talipot starches, except for that of the DH-PTS sample. The light transmittance of PTS, HM-PTS, and AC-PTS was significantly lower (*p* ≤ 0.05) than NTS. Upon phosphorylation, crosslinks formed after gelatinization changed the granular structure and made them more intact, restricted the light transmittance, and resulted in lower paste clarity [17]. Moreover, enhanced reassociation of starch chains upon HMT and autoclave treatment caused a reduction in light transmittance. Nevertheless, the light transmittance of DH-PTS was significantly higher (*p* ≤ 0.05) than PTS, and at the end of 168 h of storage, the light transmittance of DH-PTS showed a significant increase (*p* ≤ 0.05) compared to NTS. The augmentation in the paste clarity by pretreatment with DHT can be ascribed to the reduction in the amount of swollen starch granules followed by the gelling and fragmentation of amylopectin fractions. The double-helical structure and interaction of starch molecules were weakened upon DHT compared to native starch granules [36]. This induced molecules to be more dispersive and contributed to increasing paste clarity in the DH-PTS sample. Dry-heat-treated maize and sweet potato starches showed a consistent increase in light transmittance in earlier reported studies [30,36].

### 3.7. Pasting Properties

RVA profile depicting the pasting properties of native and modified talipot starch samples are shown in Figure 5. The minimum temperature needed to cook starch is known as pasting temperature (PT), and the NTS exhibited a PT of 86.51 °C. In phosphorylated and thermal pretreated phosphorylated talipot starches, the PT significantly increased (*p* ≤ 0.05) than NTS, except for the DH-PTS sample. Higher PT is due to the development of crosslinks, which provides strong intermolecular associative forces and increases the structural stability of PTS [20]. The increased PT of HM-PTS and AC-PTS indicates increased resistance of thermally modified starch granules to swelling. The additional heat uptake may be imputed to several factors, such as the development of intermolecular crosslinks, agglomeration of starch granules, and reduction in space between molecules by the fortified associative forces in the chains [31]. However, the PT of DH-PTS was found to be less than PTS as the DH-PTS sample had an unstable crystalline structure, which made the starch granules have lower resistance to swelling and rupturing and resulted in lower PT than the PTS sample [36].

Peak viscosity (PV) is the maximum viscosity shown at the equilibrium point between swelling and polymer leaching due to increased temperature, and break down viscosity (BDV) represents the starch granules’ stability against shear force and temperature. The PV and BDV of NTS were observed to be 3646 and 1881 cP. The PV of PTS, HMTS, ACTS, HM-PTS and AC-PTS showed a significant increase (*p* ≤ 0.05), whereas the DHTS and DH-PTS showed a significant decrease (*p* ≤ 0.05) compared to NTS. The increase in PV of PTS is due to the higher integration of starch granules. Modification of starch by phosphorylation enhanced the resistance of granules against disintegration by increasing the intra- and inter-bonding between starch molecules, which resulted in higher viscosity [20]. The PV of PTS was negatively correlated with its swelling power and solubility. This increased viscosity of HM-PTS and AC-PTS was ascribed to the increase in the starch granule rigidity upon hydrothermal treatments and crosslinking [37]. Besides, the reduction in PV of DHTS and DH-PTS could be attributed to the increased association between starch chains by the increase in the intra-granular bonding force [32]. Lower PV and BDV of DH-PTS indicate stronger paste characteristics by the enhanced shear and heat stability of the starch granules.

Final viscosity (FV) indicates the increase in starch viscosity because of aggregation of amylose molecules on cooling, and setback viscosity (SBV) determines the gelation or retrogradation capacity of starch [38]. The FV and SBV of NTS were found to be 2562 and 797 cP, respectively. The FV and SBV of all modified samples significantly increased (*p* ≤ 0.05) versus NTS, except for DM-PTS (Figure 5). The decrease in FV and SBV of the DM-PTS sample illustrates their lower tendency against starch retrogradation and indicates improvement in cold paste stability [36]. In HM-PTS and AC-PTS, the increase in FV and SBV was attributed to the increase in phosphorus content that enhanced inter-and intra-molecular crosslinking bonds formed with phosphorus in the modified starch chains that help in the formation of additional junction zones in the continuous phase of starch gels [35]. Besides, the increase in FV and SBV may also be due to the realignment of amylose units leached through the fissures formed on the surface of talipot starch granules upon hydrothermal treatments (Figure 1). High FV of modified talipot starches relates their suitability in the application that uses high-temperature processing or requires high viscosity in the final products [20].

### 3.8. Textural Properties

The textural parameters of native and modified talipot starches are presented in Table 2. Textural parameters of single- and dual-modified starches showed a significant difference (*p* ≤ 0.05) from NTS. Gel hardness has been reported to be dependent on the binding capacity of starch molecules to water by hydrogen bonding [39]. Adhesiveness is the energy needed to overcome the attractive force between starch gel and the surface with which gel is in contact, and which is the negative force area of TPA graph. Cohesiveness determines the internal bond strength of the starch gel, and it indicates the ability of the gel to hold together, while springiness indicates the elastic property of the starch gel [40]. The hardness, adhesiveness, springiness, cohesiveness, and gumminess of NTS was observed to be 45.72 N, −16.26 Nmm, 0.90 mm, 0.46, 21.03 Nmm, respectively. Phosphorylation and thermal treatments significantly enhanced the textural properties of PTS gels. The increase in hardness of PTS could be attributed to the inter- and intra-molecular bonding due to crosslinking, which strengthened the gel structure and prevented the rupture during compression [20].

The improved gel properties in DHTS might be due to the molecular depolymerization promoted by DHT that enhances the reassociation and packing of starch chains, which helps in developing a stronger three-dimensional network structure [8]. In this way, DHT promoted the formation of stronger gels, expanding the industrial applications of talipot starch. Lima et al. [41], in their study with cassava starch upon DHT, developed hydrogels possessing textural properties suitable for the potential application in 3D printing foods. In HMTS and ACTS, the increase in the hardness of gels is attributed to the increased crosslinking between starch chains in the particular amylose portion, resulting in the formation of more junction zones in the continuous phase of the gel that subsequently increases the gel hardness [39]. The increase in gel hardness of modified starches can be ascribed to an increased rate of retrogradation, resulting in the formation of more rigid granules. The dual-modified starches showed higher firmness than the native and single-modified starches, which might be due to the combined effect of both treatments. Higher values of firmness and cohesiveness of the crosslinked and dual-modified starch pastes might be attributed to the increase in granular integrity of the modified starches due to phosphorylation. Phosphorylated starches might be used to control the texture of products as it is able to tolerate heat, shear and acid during processing [20].

### 3.9. Dynamic Rheology

The rheological properties dictate the flow and deformation behavior of starch under stress. The rheological properties of the starch gels are crucial to determine the processability and eating quality of starch-based food products. The G′ and G″ of native and modified talipot starch samples over an angular frequency (ω) of 1–100 rad/s are depicted in Figure 6a. G′ indicates the elastic nature of the starch gel, and it is the amount of energy recovered after each deformation cycle, while Gʺ represents the viscous nature of the starch gel, and it is the amount of energy lost as viscous dissipation after each deformation cycle [25]. Native and all modified samples exhibited G′ > G″ upon rheological analysis, revealing the elastic property of the starch. Among the modified starches, AT-PTS showed the highest G′ and G″ values. Upon modification, the G′ and G″ increased due to the increased crosslinking between the starch chains and facilitated the development of junction zones in the continuous phase of starch gels. This results in an augmentation in the gel strength after phosphorylation. In HMTS and ACTS samples, the increased G′ was ascribed to the enhanced retrogradation tendency by hydrothermal treatments, which aids in increased bonding between amylose and amylopectin. Besides, pretreatment with DHT, HMT, and autoclaving increased firmness and gel strength [42]. Compared to NTS, all phosphorylated talipot starch gels showed higher paste stability and formed stronger gels resistant to excessive heat and shear due to decreased amylose content and swelling power. Similarly, Li et al. [43], in their work, stated that the gel formation of waxy rice starch was enhanced with the addition of xanthan along with DHT. Hence, in our study with phosphorylation, the association formed between phosphate groups and starch made stronger gels that are stable to shearing and heating. Such starch pastes or gels that require higher energy for sheeting or rolling are suitable for the preparation of flattened bread, pasta and noodles [21].

Tan δ value is the ratio of G″ and G′, where tan δ value > 1 implies the viscous behavior of starch gels, and tan δ value < 1 implies the elastic behavior of starch gel. The tan δ values of NTS, crosslinked and thermally modified talipot starches, were below unity (Figure 6b). This indicated the elastic behavior or solid-like characteristics of talipot starch gels. Thermal treatment and phosphorylation synergistically made the talipot starch gels more stable under mechanical and thermal stress [43]. Therefore, modified talipot starch gels with greater rigidity and improved capacity of recovering their shape after enduring stress will be suitable for 3D printing applications [8].

### 3.10. In Vitro Digestibility

Based on gastrointestinal absorption, starch is classified into RDS, SDS, and RS. The NTS after hydrolysis showed the RDS, SDS, and RS as 29.44%, 33.85%, and 36.71%, respectively. RS possesses varied physiological benefits by acting as a dietary fiber and reduces the risk of many degenerative diseases. From Table 3, it is evident that phosphorylation and thermal treatments had a noticeable influence on the digestibility of talipot starch. Upon phosphorylation, the in vitro digestibility significantly reduced (*p* ≤ 0.05) for talipot starch. The RS and SDS content increased to 44.37% and 35.01% upon crosslinking with POCl_3_, owing to the restriction of swelling by the starch granules and enhanced resistance to digestion by α-amylases. Crosslinked waxy maize starch with STMP/STPP, 99:1 (5, 10, and 12%, *w*/*w*) showed a consistent increase in RS content [17]. The DHTS and DH-PTS sample showed increased digestibility with a significant increase (*p* ≤ 0.05) in RDS content versus PTS. This might be due to the partial disruption of organized chain structures by DHT, which could promote the attack of the enzymes into the interiors of starch granules [44]. Hence, structural disintegration induced by the DHT was not able to increase the RS content of talipot starch with phosphorylation compared to phosphorylation treatment alone. Liu et al. [44], in their study with waxy potato starch, also revealed a consistent increase in starch digestibility upon DHT.

HMTS and ACTS showed an increased percentage of RS (resistant starch type III, RS3) compared to NTS by partial hydrolysis. RS3 is a type of RS formed via retrogradation mechanism by intentional modification or processing. The effect of hydrothermal treatment on in vitro digestibility is governed by starch source, moisture content during treatment, treatment time and temperature, and interactions between amylose–amylose, amylose–amylopectin, amylopectin–amylopectin interaction, and amylose–lipid interactions [45]. However, the thermal pretreatment with HMT and autoclaving prior phosphorylation remarkably decreased the digestibility of talipot starches compared to PTS. Soler et al. [13], in their study, stated that corn starch upon autoclave treatment had repercussions for RS formation that possess extended application in processibility and nutraceutical properties of starch. The significant increase (*p* ≤ 0.05) in the RS content of AC-PTS is due to the combined effect of thermal force and pressure given by autoclave treatment and phosphodiester bond crosslinks. Upon hydrothermal treatments, the tight structure in the amorphous regions formed by starch granule expansion might have reduced enzyme susceptibility in AC-PTS for in vitro digestion [34]. Hence, combined modification of starch with HMT or autoclaving with phosphorylation helps to significantly improve the RS content, which possesses several health benefits such as hypocholesterolemic and hypoglycemic effects, probiotic activity, maintaining colon health, and preventing gall-stone formation [45].

## 4. Conclusions

Talipot starch modified by phosphorylation alone and phosphorylation and thermal pretreatments remarkably changed the starch granule characteristics and starch paste functionality compared to NTS. The phosphorylation modification in talipot starch was confirmed by the increased P content and DC of PTS versus NTS, and the P content and DC of thermal pretreated phosphorylated starches was significantly greater (*p* ≤ 0.05) than PTS. This increase in P content and DC is in agreement with the increased peak intensity of 1160 and 1010 cm^−1^ representing P=O and P–O–C stretching vibrations upon phosphorylation. The swelling power of PTS, DH-PTS, HM-PTS and AC-PTS was significantly lower than (*p* ≤ 0.05) NTS, due to reduced amylose content and relative crystallinity by the introduction of phosphodiester bonds upon crosslinking. The increased light transmittance of NTS and DH-PTS suggested its application in the preparation of soups, jellies, and pie fillings that requires higher paste clarity. The increased PT, PV and FV of HM-PTS and AC-PTS indicate the development of intermolecular crosslinks and reduced space between the molecules by fortified associative forces in starch chains. Such starches are suitable for application, which uses high-temperature processing or requires higher viscosity in the final products. The increased G′ and G″ values of DH-PTS, HM-PTS and AC-PTS depict that stronger gels are formed upon dual modification that is stable to shearing and heating. In vitro digestibility significantly increased (*p* ≤ 0.05) with DH-PTS sample and significantly decreased (*p* ≤ 0.05) with PTS, HM-PTS and AC-PTS. The improved RS content of HM-PTS and AC-PTS suggests its application in the preparation of low glycemic food products. Talipot starch crosslinked with phosphorylation with three different thermal modifications as pretreatments distinctly changed starch properties and indicated their functionality in varied food applications.

## Figures and Tables

**Figure 1 polymers-13-03855-f001:**
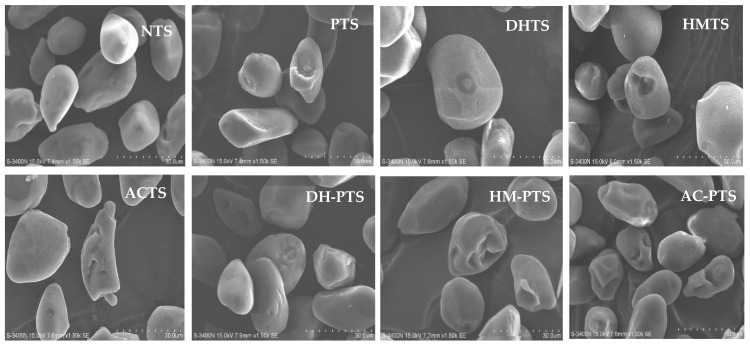
Scanning electron micrographs of native, single and dual-modified talipot starch samples.

**Figure 2 polymers-13-03855-f002:**
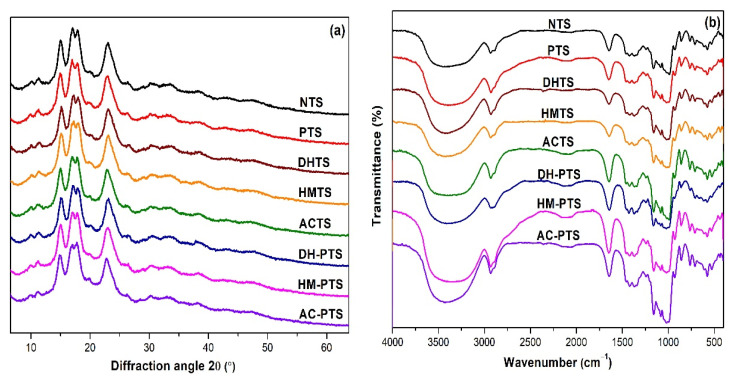
(**a**) X-ray diffractograms. (**b**) FT-IR spectra of native, single-, and dual-modified talipot starch samples.

**Figure 3 polymers-13-03855-f003:**
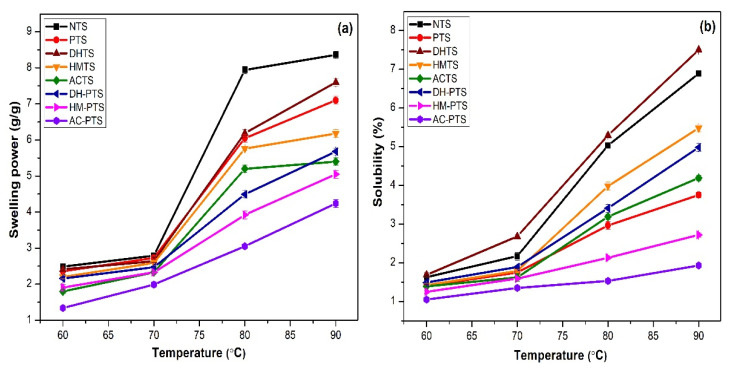
(**a**) Swelling power. (**b**) Solubility of native, single-, and dual-modified talipot starch samples.

**Figure 4 polymers-13-03855-f004:**
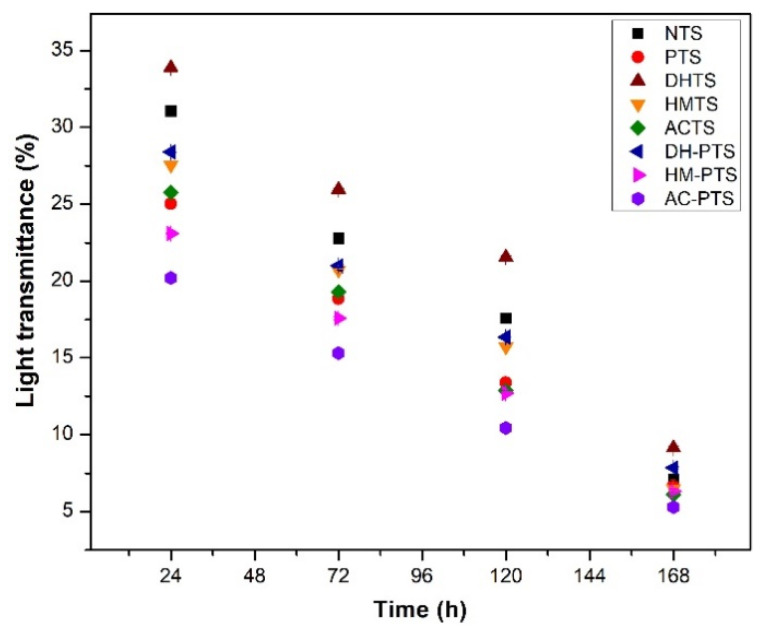
Light transmittance of native, single-, and dual-modified talipot starch samples.

**Figure 5 polymers-13-03855-f005:**
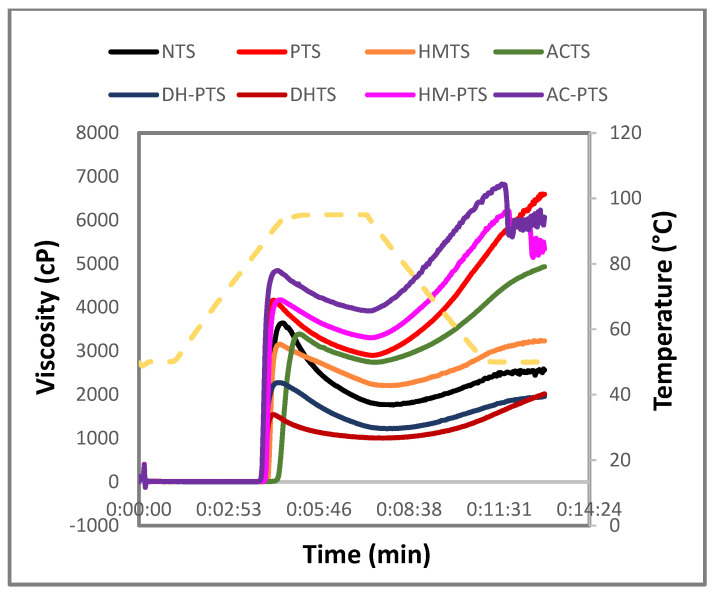
RVA profile of native, single-, and dual-modified talipot starch samples.

**Figure 6 polymers-13-03855-f006:**
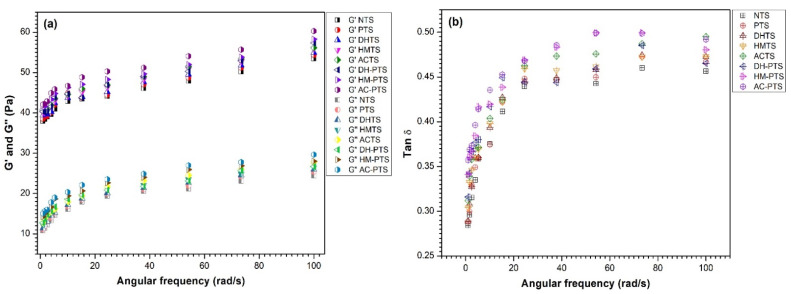
(**a**) Storage and loss moduli. (**b**) Loss factor of native, single-, and dual-modified talipot starch samples.

**Table 1 polymers-13-03855-t001:** Phosphorous content (P), degree of crosslinking (DC), amylose content and relative crystallinity (RC) of native, single- and dual-modified talipot starch samples.

Samples	P (%)	DC	Amylose (%)	RC (%)
NTS	0.018 ± 0.03 ^a^	–	28.81 ± 0.01 ^f^	16.97 ± 0.02 ^d^
PTS	0.058 ± 0.03 ^b^	0.00298 ^a^	25.23 ± 0.03 ^e^	16.70 ± 0.10 ^d^
DHTS	0.018 ± 0.02 ^a^	–	23.39 ± 0.08 ^d^	13.57 ± 0.10 ^c^
HMTS	0.016 ± 0.07 ^a^	–	25.19 ± 0.17 ^e^	15.82 ± 0.20 ^f^
ACTS	0.016 ± 0.10 ^a^	–	31.02 ± 0.10 ^g^	14.20 ± 0.03 ^e^
DH-PTS	0.069 ± 0.05 ^c^	0.00361 ^b^	19.26 ± 0.10 ^a^	10.60 ± 0.02 ^a^
HM-PTS	0.076 ± 0.05 ^d^	0.00398 ^c^	20.54 ± 0.06 ^b^	13.51 ± 0.04 ^c^
AC-PTS	0.080 ± 0.01 ^e^	0.00418 ^d^	22.75 ± 0.11 ^c^	12.12 ± 0.01 ^b^

Values articulated are the mean of triplicate ± SD. The values within the same column with different superscript alphabets (^a^–^g^) indicate significant difference of *p* ≤ 0.05.

**Table 2 polymers-13-03855-t002:** Textural parameters of native, single-, and dual-modified talipot starch samples.

Samples	Hardness (N)	Adhesiveness (Nmm)	Springiness (mm)	Cohesiveness	Gumminess (Nmm)
NTS	45.72 ± 0.03 ^a^	−16.26 ± 0.10 ^a^	0.90 ± 0.10 ^a^	0.46 ± 0.11 ^a^	21.03 ± 0.09 ^a^
PTS	82.42 ± 0.12 ^e^	−11.38 ± 0.20 ^e^	0.96 ± 0.03 ^d^	0.54 ± 0.04 ^d^	44.50 ± 0.03 ^e^
DHTS	51.72 ± 0.02 ^b^	−15.23 ± 0.12 ^b^	0.92 ± 0.12 ^b^	0.52 ± 0.11 ^b^	26.89 ± 0.30 ^b^
HMTS	55.29 ± 0.09 ^c^	−15.21 ± 0.11 ^c^	0.93 ± 0.11 ^c^	0.53 ± 0.10 ^c^	29.30 ± 0.09 ^c^
ACTS	68.26 ± 0.10 ^d^	−13.97 ± 0.02 ^d^	0.96 ± 0.20 ^d^	0.56 ± 0.02 ^e^	38.22 ± 0.03 ^d^
DH-PTS	90.83 ± 0.11 ^f^	−9.38 ± 0.07 ^f^	1.00 ± 0.09 ^e^	0.62 ± 0.19 ^f^	56.32 ± 0.10 ^f^
HM-PTS	97.29 ± 0.18 ^g^	−8.49 ± 0.20 ^g^	1.00 ± 0.01 ^e^	0.63 ± 0.13 ^g^	61.29 ± 0.15 ^g^
AC-PTS	113.14 ± 0.11 ^h^	−7.62 ± 0.08 ^h^	1.03 ± 0.06 ^f^	0.66 ± 0.01 ^h^	74.67 ± 0.03 ^h^

Values articulated are the mean of triplicate ± SD. The values within the same column with different superscript alphabets (^a^–^h^) indicate significant difference of *p* ≤ 0.05.

**Table 3 polymers-13-03855-t003:** In vitro digestibility of native, single-, and dual-modified talipot starch samples.

Samples	RDS (%)	SDS (%)	RS (%)
NTS	29.44 ± 0.05 ^f^	33.85 ± 0.11 ^d^	36.71 ± 0.02 ^b^
PTS	20.62 ± 0.14 ^b^	35.01 ± 0.31 ^e^	44.37 ± 0.13 ^e^
DHTS	35.20 ± 0.03 ^g^	30.65 ± 0.08 ^a^	34.15 ± 0.09 ^a^
HMTS	27.71 ± 0.16 ^e^	33.31 ± 0.02 ^c^	38.98 ± 0.18 ^c^
ACTS	22.54 ± 0.11 ^d^	35.38 ± 0.02 ^e^	42.08 ± 0.15 ^d^
DH-PTS	27.93 ± 0.12 ^e^	32.97 ± 0.27 ^b^	39.10 ± 0.10 ^c^
HM-PTS	21.70 ± 0.02 ^c^	33.86 ± 0.10 ^d^	44.44 ± 0.05 ^e^
AC-PTS	15.91 ± 0.17 ^a^	36.01 ± 0.18 ^f^	48.08 ± 0.03 ^f^

Values articulated are the mean of triplicate ± SD. The values within the same column with different superscript alphabets (^a^–^g^) indicate significant difference of *p* ≤ 0.05.

## Data Availability

The data presented in this study are available on request from the corresponding author.

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
