# Peer review of "Effect of Thermal Pretreatments on Phosphorylation of Corypha umbraculifera L. Stem Pith Starch: A Comparative Study Using Dry-Heat, Heat-Moisture and Autoclave Treatments"

_polymers, 2021, doi:10.3390/polym13213855_

Round 1

Reviewer 1 Report

In this work, the authors studied the effect of different thermal pretreatments and phosphorylation on physical properties, viscoelastic behavior, and digestibility of talipot starch. The manuscript is well written, and the subject is current and important for food industry and beyond. This manuscript is recommended to be published after including and addressing the below listed comments with major corrections.

- The authors should clearly explain the influence of different thermal treatments (DHT, HMT, and AC) on properties of starch in the introduction.

- The authors should write the name and information of the chemicals used for this study in the materials part of this manuscript.

- How much was the concentration of starch in the gels for dynamic viscosity? Was it also 12%?

- This is a comparative study, and the objective of this study is to examine the effect of thermal pretreatments on properties of talipot starch. However, the effect of different thermal treatments on swelling and solubility of starch is not clear. It’s worthy if the authors study and discuss the effect of thermal treatments on talipot starch without a phosphorylation treatment.

-The authors should improve the discussion of the manuscript. The role and influence of hydroxyl groups, RC content, and amylose content on swelling power, solubility, and rheology are overlooked.

- It is worthy if the authors examine the viscosity of starch gels as a function of shear rate and perform an amplitude sweep test to study the flow behavior and microstructure of the starch gels.

- Please double check all the abbreviations in the manuscript. For example: line 406, 413, 421 DM-PTS.

- The quality of the graphs should be improved. Please use symbols with different shapes and styles to present each item.

Reviewer 2 Report

Detailed recommendation:

Abstract: please add more data to the abstract.

Key words: add: chemical modification

Introduction: In the introduction, please add more information about the properties of product after starch phosphorylation

In how many repetitions did you make your analytical methods?

Why you didn’t make textural properties of starch gels? I think this is important

Round 2

Reviewer 1 Report

Thanks for the corrections. The manuscript is ready for publication.